# Improving Math Problem Solving in Large Language Models Through Categorization and Strategy Tailoring

### Abstract

In this paper, we investigate how to harness large language models (LLMs) to solve mathematical problems both quickly and accurately. Specifically, we demonstrate the effectiveness of classifying problems into distinct categories and applying category-specific problem-solving strategies to enhance the math performance of LLMs. We develop a straightforward machine learning model for problem categorization and show that its accuracy can be significantly improved through the creation of well-designed training datasets. We believe that our approach works by helping reduce hallucinations in LLMs, which is a critical step toward unlocking their potential to tackle advanced mathematical problems.

## Introduction

Large language models (LLMs) have significantly influenced numerous domains, including finance (Nie et al. 2024), scientific writing (AI4Science and Quantum 2023; Geng et al. 2024), and law (Colombo et al. 2024). Recently, mathematics has emerged as a challenging new frontier for LLMs (Srivatsa and Kochmar 2024). Solving mathematical problems requires more than mechanistic computation; it often demands non-obvious insights and creative approaches. Unlike many tasks that can be framed as variations of next-token prediction, mathematical problem-solving involves generating unique strategies and uncovering subtle insights, making it inherently difficult for machine learning models.

Unsurprisingly, many AI models struggle with mathematical problems, frequently producing inaccurate results. For instance, even widely known models like ChatGPT often fail to solve simple competition math problems correctly (Greenleaf 2023). One primary reason is hallucination—where the model generates responses based on faulty logic or incorrect assumptions.

To address these challenges, researchers at Google DeepMind have developed models like AlphaGeometry and AlphaProof, which integrate text generation with formal reasoning and theorem-proving techniques (Roberts 2024). These models have shown promising results: for example, AlphaProof achieved a score of 28 on the 2024 International Mathematics Olympiad (IMO), earning a silver medal (Wilkins 2024), while AlphaGeometry outperformed the average IMO silver medalist on geometry prob-

lems (Trinh and Luong 2024). However, these models have limitations. AlphaProof, for instance, can take up to three days to solve some IMO problems (Wilkins 2024), far exceeding competition time limits and making it impractical for real-time solutions. Additionally, AlphaGeometry is restricted to geometry problems, and AlphaProof performs poorly on combinatorics tasks (Williams 2024).

This paper focuses on the specific but significant challenge of enabling LLMs to compute solutions to mathematical problems both *quickly* and *accurately*. Unlike AlphaProof and AlphaGeometry, our approach targets computational problems that require producing numeric answers. Even within this narrower scope, hallucinations in LLM outputs can lead to particularly egregious errors.

We propose a novel approach that helps mitigate hallucination by *categorizing problems into predefined categories and providing category-specific solving strategies as input to the LLM*. Our framework identifies four primary categories—algebra, combinatorics, geometry, and number theory—but can be extended to accommodate more nuanced categorizations. Depending on the problem's category, we employ one of two problem-solving strategies: chain of thought or program of thought. This structured methodology substantially improves the accuracy of LLM problem-solving.

To automate problem categorization, we developed a lightweight deep neural network model. The key to its effectiveness lies in curating *the right training data*. In particular, we show the importance of domain-specific factors such as "answer extraction" and achieving a good balance of different problem categories. We then show how to probabilistically associate a problem-solving strategy with each identified category.

Our empirical analysis demonstrates that our categorization model achieves over 80% accuracy. Furthermore, we evaluate the impact of our neural model-categorization-driven strategy selection against two baselines: using ground-truth categories for strategy selection and randomly assigning strategies. Our approach outperforms random strategy assignment by 67% but falls 29% short of the performance achieved using ground-truth categories.

To summarize, our contributions are as follows:

1. We find that prompting LLMs with problem categories and associated problem-solving strategies significantly

improves problem-solving accuracy.

2. We develop a lightweight neural model for problem categorization.

3. We show the importance of - and the method for - obtaining the right training data for effective neural categorization.

4. We empirically demonstrate the problem-solving accuracy of our approach and compare it against key baselines.

## Background

**Hallucination** occurs when an LLM produces reasoning that is incorrect or generates information that is not necessarily true. This issue is particularly critical in mathematical problem-solving. If an LLM hallucinates and arrives at an incorrect intermediate result inconsistent with the given information, it is highly unlikely to recover and successfully solve the problem.

Our overall approach enhances LLMs' reliability by introducing category-specific math problem-solving strategies. These strategies help mitigate hallucinations by providing the model with more structured and precise prompts, offering greater context to guide the solution process. In particular, we use two strategies: Chain of Thought (CT) (Wei et al. 2022) and Program of Thought (PT) (Chen et al. 2023).

**Chain of Thought** (Wei et al. 2022) involves asking an LLM to come up with a detailed, step-by-step solution to a problem before returning the final answer. This approach encourages logical reasoning, which helps reduce errors and hallucinations in problem-solving.

**Program of Thought** (Chen et al. 2023) involves instructing the LLM to create a detailed Sympy-based Python script to solve the problem and then execute the code to obtain the answer. Sympy is a Python library for symbolic mathematics that can solve equations with multiple variables and perform various algebraic operations. If the generated code produces an incorrect result, the model iteratively refines and re-runs the code until the correct solution is found.

While CT can be applied broadly to any mathematical problem to mitigate hallucination, it is particularly effective for problems requiring careful reasoning. On the other hand, PT is better suited for problems involving iteration or recursion, as it can execute such operations efficiently and accurately.

We integrate both CT and PT strategies with an LLM called Deepseek-Math (Shao et al. 2024). Deepseek-Math is a compact LLM fine-tuned for solving mathematical problems, chosen for its compatibility with low-end GPUs commonly found in online Jupyter Notebooks. This accessibility makes it especially valuable for math students and educators worldwide.

Next, we illustrate the effectiveness of CT and PT through specific examples. CT is very effective for the following problem:

**Problem 1** *What is the minimum value of $5x^2 + 5y^2 - 8xy$ when $x$ and $y$ range over all real numbers such that $|x - 2y| + |y - 2x| = 40$?*

This problem can be solved by CT by noting that $5x^2 + 5y^2 - 8xy = (x - 2y)^2 + (y - 2x)^2$. After this inference, there are multiple ways to solve the problem, which is now drastically easier.

PT is very effective for the following problem:

**Problem 2** *Suppose that we roll four 6-sided fair dice with faces numbered 1 to 6. Let $a/b$ be the probability that the highest roll is a 5, where $a$ and $b$ are relatively prime positive integers. Find $a + b$.*

One possible solution with PT would be to iterate through every single possible roll of the dice and count the number of rolls with the highest number 5. Because each roll is equally likely, this allows the model to easily compute the total probability. Additionally, this algorithm would run very fast because the number of cases is just $6^4 = 1296$, which can be covered by a computer in a fraction of a second.

One naive approach to solving a math problem is to randomly try CT or PT. Should the selected strategy be inappropriate, not only is the obtained answer likely to be incorrect but significant time is expended in arriving at it. This is a poor fit for our setting where we aim to solve math problems correctly and quickly.

## Related Work

In this section, we discuss recent works in the general area of using AI techniques and LLMs to solve mathematics problems. We draw the reader's attention to a recent thorough survey (Ahn et al. 2024) that covers much of the progress made in this field, particularly highlighting the development of math-specific fine-tuned models derived from state-of-the-art LLMs.

Several studies, such as (Kao, Wang, and Hsieh 2024) and (He-Yueya et al. 2023), focus specifically on solving algebra problems using LLMs. For example, (Kao, Wang, and Hsieh 2024) addresses algebraic problems involving two or more variables, while (He-Yueya et al. 2023) explores the use of symbolic solvers for tackling algebraic word problems. In contrast, our work differs in two key ways: (a) it spans multiple problem categories, and (b) it emphasizes problems from mathematics competitions, which are succinctly specified and present unique challenges.

A related approach involves using one LLM to solve problems while employing another LLM to evaluate solutions or provide feedback and hints, as discussed in (Agrawal et al. 2024) and (Wu et al. 2024). Unlike these methods that provide generic hints, our approach provides highly specific guidance by categorizing problems and instructing the LLM on which strategy to use based on the problem's category.

Other works explore complementary aspects of problem-solving. For instance, (Xu et al. 2024) examines how the length of word problems impacts LLM performance, while (Anantheswaran et al. 2024) investigates the effects of irrelevant information on problem-solving accuracy and proposes methods to mitigate this issue. These techniques could synergize with our approach, potentially enhancing accuracy further: for example, it would be possible to integrate measures to reduce noise and reduce problem length in prompts given to the model.

## Overview of Our Approach

Our method extracts relevant information about a problem before the model attempts to solve it. Using this information, we select the most suitable strategy between Program of Thought (PT) and Chain of Thought (CT) for solving the problem. This approach not only increases the likelihood of arriving at the correct solution but also reduces the number of attempts required to reach an answer.

Our overall approach to solving problems is as follows:

1. Determine the *category* of a problem, choosing from one of four: algebra, combinatorics, geometry, or number theory.

2. Use this to determine the *probability distribution* for picking one of {PT, CT} as the strategy for this problem.

3. Use this chosen strategy to solve the problem.

4. Repeat steps 2 and 3 until some answer has the desired frequency or a problem has been attempted a fixed number of times. In either case, conclude that the most frequent answer is the final answer.

The four categories—algebra, combinatorics, geometry, and number theory—are chosen because they align with the structure of problems in many major international math competitions, such as the International Mathematics Olympiad (IMO). Problems within the same category often share similar characteristics, making this categorization highly relevant for strategy selection.

In Section , we empirically demonstrate how categorizing problems enhances LLM problem-solving. Prior to that, in Section , we discuss the rationale behind step 2, including why certain strategies are more effective for specific problem types and the benefit of employing a probability-based approach for strategy selection.

### On Using Different Strategies

Typically, geometry problems will be easier to solve using CT – because they do not involve multiple cases, PT (typically involving iteration) will not benefit problem-solving. On the other hand, combinatorics problems will typically be easier to solve using PT, because they often involve counting cases. (This is true with counting and probability problems.)

However, a notable exception to the above observation is the following problem:

**Problem 3** *Let the 'sparkle' operation on positive integer $n$ consist of calculating the sum of the digits of $n$ and taking its factorial, e.g. the sparkle of 13 is $4! = 24$. A robot starts with a positive integer on a blackboard, then after each second for the rest of eternity, replaces the number on the board with its sparkle. For some 'special' numbers, if they're the first number, then eventually every number that appears will be less than 6. How many such special numbers are there with at most 36 digits?*

The main idea of this problem is that the sum of the digits of any special number must be at most 2. For this problem, a PT approach without any nontrivial observations would involve iterating through $10^{36}$ possibilities for our number $n$. The PT procedure outlined above involves solving every part

of the problem using code rather than making inferences, so this procedure would likely not work.

Thus, *even though problems in some categories may usually be more amenable to one strategy than the other, both strategies should still be considered*, as we do in Step 2.

Problems in algebra and number theory are equally amenable to both CT and PT because while Sympy can iterate through integers in a range quickly and can solve multivariate equations, it will have trouble solving complicated, layered problems. Thus, we can choose either strategy with equal probability.

We conclude this section by stating the weights that our approach will use in the probability distribution for choosing between CT and PT. For both algebra and number theoretic problems, we set the probability for each of CT and PT to be 50% based on the discussion above. For geometry problems, we use a weight of 90% for CT and 10% for PT. For combinatorics problems, we use a weight of 35% for CT and 65% for PT. In the latter two cases, we found that the chosen probabilities offered the empirical best performance for each category.

### How Useful is Categorization?

We now demonstrate the utility of using categorization to improve problem-solving accuracy. Specifically, we compare the performance of Deepseek-Math in two scenarios: 1. **Category-Based Strategy Selection:** Deepseek-Math is provided with the strategy corresponding to the *correct* category of the problem, which we determine through manual labeling. Based on this categorization, a strategy from {CT, PT} is selected using the defined probability distribution, and the model is tasked with solving the problem using the selected strategy. 2. **Random Strategy Selection**: Deepseek-Math is given a randomly selected strategy from {CT, PT} for each problem, without any regard for categorization.

In both scenarios, the model is allowed up to five attempts per problem; in each attempt the strategy is reselected from the corresponding distribution. This comparison highlights the effectiveness of leveraging problem categorization to guide strategy selection.

The dataset we use for this experiment consists of 25 problems similar to those from the AIME as well as problems drawn directly from the 2015 AIME I (which were not used in any training data). The results are shown in Table 1. The results from the first situation are shown in blue, and those from the second are shown in brown.

We make the following observations:

1. When using category-based choice between CT and PT, Deepseek-math solves 28% of the problems (7 out of 25), versus only 12% (3 out of 25) of the problems when using random choice.

2. When using category-based choice, Deepseek-math has a nonzero correct answer frequency on 36% of the problems (9 out of 25), versus 24% of the problems for random choice (6 out of 25).

3. When using category-based choice, Deepseek-math solves every problem that it solves using random choice.

| Problem | Outputs | Frequencies | Correct Answer | Correct Answer Frequencies | Correct? |
|---|---|---|---|---|---|
| 1 | 52 30 | 2 2 | 52 | 2 0 | Yes No |
| 2 | 891 445 | 2 2 | 250 | 0 0 | No No |
| 3 | 1 14 | 2 1 | 702 | 0 0 | No No |
| 4 | 0 2 | 2 2 | 800 | 1 1 | No No |
| 5 | 310 10 | 2 2 | 211 | 0 0 | No No |
| 6 | 100 1 | 2 2 | 199 | 1 1 | No No |
| 7 | 97 97 | 2 4 | 185 | 0 0 | No No |
| 8 | 256 256 | 2 2 | 320 | 0 0 | No No |
| 9 | 464 229 | 2 1 | 480 | 0 0 | No No |
| 10 | 199 793 | 2 2 | 199 | 2 1 | Yes No |
| 11 | 39 750 | 1 2 | 722 | 0 0 | No No |
| 12 | 139 139 | 3 4 | 139 | 3 4 | Yes Yes |
| 13 | 307 307 | 2 2 | 307 | 2 2 | Yes Yes |
| 14 | 768 800 | 2 1 | 507 | 0 0 | No No |
| 15 | 4 316 | 2 2 | 341 | 0 0 | No No |
| 16 | 12 150 | 1 2 | 58 | 0 0 | No No |
| 17 | 396 396 | 2 2 | 539 | 0 0 | No No |
| 18 | 695 695 | 4 2 | 695 | 4 2 | Yes Yes |
| 19 | 494 604 | 2 2 | 494 | 2 0 | Yes No |
| 20 | 72 12 | 2 3 | 72 | 2 0 | Yes No |
| 21 | 24 45 | 1 2 | 108 | 0 0 | No No |
| 22 | 143 509 | 2 2 | 431 | 0 0 | No No |
| 23 | 14 1 | 1 2 | 91 | 0 0 | No No |
| 24 | 999 10 | 2 1 | 483 | 0 0 | No No |
| 25 | 2 85 | 1 2 | 53 | 0 0 | No No |

Table 1: Outputs of Deepseek-math while using category-based choice between CT and PT. The second and third columns are the most frequent outputs for each problem. The blue table entries correspond to the model using a strategy based on the correct category, whereas the brown entries correspond to choosing randomly between CT and PT.

4. The average output frequency, in either case, is similar (1.92 vs 2.04 in category-based choice versus random choice). However, the average correct output frequency of category-based choice is 0.76, versus 0.44 for random choice.

In summary, our empirical analysis shows that categorization is useful for facilitating LLM problem-solving.

## Categorization Model

In this section, we first build a categorization model. We then study the accuracy of the model and improve on it in later sections.

Our model is a deep neural network with three layers, whose architecture we will describe in detail shortly. To train the model, we use a 412-problem subset of the MATH dataset (Hendrycks et al. 2021), a collection of problems written in LaTeX and categorized into five difficulty levels. We focus specifically on level 5 problems because they are comparable in difficulty to those found in AMC 12 and AIME competitions, whereas the other levels are significantly easier.

The MATH dataset categorizes problems into seven groups, three more than the four categories (algebra, geometry, number theory, and combinatorics) used in our approach. These additional categories are prealgebra, precalculus, and intermediate algebra.

We exclude problems from the prealgebra category because they are not only much simpler than AIME-level problems but also broadly distributed across the other categories (e.g., algebra, geometry, number theory, and combinatorics). Similarly, problems in the precalculus and intermediate algebra categories are primarily solved using algebraic techniques, so we classify them as algebra problems in our framework.

One very effective way to predict the category of a problem is to use *word frequency*. This is because some words can be considered 'indicators', or words that indicate that a problem is very likely to be of a particular . Table 2 shows examples of several indicators found in the MATH dataset:

Some standard indicators can be easily identified, such as the words *polynomial*, *probability*, and *integer*. However, seemingly arbitrary words and code commands can also serve as indicators for specific categories due to their contextual significance. For instance, the syntax '++i);' is commonly used to create iterative diagrams in Asymptote, a vector graphics language predominantly used in combinatorics problems. Thus, this syntax becomes a meaningful indicator for the combinatorics category.

Focusing on our neural model, it consists of three layers. The first layer includes one node for each indicator, the second layer contains ten hidden nodes, and the third layer has four output nodes, each representing one of the cate-

| | Indicator |
|---|---|
| Algebra | Polynomial |
| Algebra | Complex |
| Algebra | \dots |
| Algebra | \begin{array}{cl} |
| Combinatorics | Committee |
| Combinatorics | Probability |
| Combinatorics | ++i); |
| Combinatorics | $(a,b,c)$ |
| Geometry | Tangent |
| Geometry | Rectangle |
| Geometry | ['asy'] |
| Geometry | label($a$ |
| Number Theory | Integer |
| Number Theory | Remainder |
| Number Theory | \cdots |
| Number Theory | $n$ |

Table 2: Some sample indicators for each

gories: algebra, combinatorics, geometry, or number theory. The model employs a sigmoid activation function to account for the possibility that some problems may span multiple categories.

Before training the model, it is essential to address the imbalance in the sizes of each category within the dataset. Specifically, the training data consists of 237 algebra problems, 44 combinatorics problems, 40 geometry problems, and 91 number theory problems. If the model were trained directly on this data, its predictions would be heavily biased towards algebra. In fact, when we tested this, the model assigned a 93% probability to the algebra category when provided with an empty string as input.

To mitigate this issue, we balanced the dataset by duplicating the non-algebra problems so that all categories are approximately equally represented.

This model achieves between a 60% and 70% accuracy with the categorization of problems around the difficulty level of the AIME (see Section  for suitable results). We now examine how to further improve the model's accuracy.

## The Right Data Matters!

To improve our accuracy, we examine one of the problems that our model consistently gets wrong:

**Problem 4** *Suppose that we roll four 6-sided fair dice with faces numbered 1 to 6. Let $a/b$ be the probability that the highest roll is a 5, where $a$ and $b$ are relatively prime positive integers. Find $a + b$.*

The model evaluates the category of this problem as number theory, whereas in reality, the category is combinatorics. If we search for specific number-theoretic keywords in this problem, the phrase "relatively prime positive integers" will jump out. While this is a number-theoretic concept, it is not part of the real substance of the problem, rather it is *a method of extracting an integer answer from the fraction $a/b$*. This part of a problem is generally known as the 'answer extraction' of a problem: it is the part of a problem that consists

of work and calculations done after the main quantities in a problem are found. Answer extraction appears frequently on contests such as the AIME, which has a specific answer format.

One issue is that the MATH dataset has a much lower proportion of problems with answer extraction compared to what appears on the AIME and the AMC 12.

Another issue here is that most problems with answer extractions will be categorized as number theory by our model. This is because answer extractions typically have many number theory indicators. For example, one of the most common answer extractions on the AIME asks the competitor to first represent a rational quantity as $\frac{x}{y}$ for *relatively prime positive integers* $x$ and $y$ and then compute the sum $x+y$. The italicized text in the previous section contains four keywords which are likely number theory indicators. Other problems in competitions such as the AIME involve similar number-theoretic answer extraction methods.

What we notice here is that a key reason for the low accuracy of our model is the nature of the data used for training. We address this by adding more problems to our training dataset such that the proportion of problems that involve answer extraction methods is comparable to AIME. Training using this richer data is helpful – it enables our model to experience problems that involve number theory indicators but are *not actually categorized as number theory*. This helps our model avoid categorizing problems with answer extractions as always being number-theoretic.

The best option for problems to add to our dataset is problems from the AIME itself. These problems satisfy the aforementioned constraints and are very similar to the problems that we would ultimately like to solve. Hence, we hand-label and add every problem from the last six AIME contests – totaling 90 problems – to our training dataset. We will see in the next section that the addition of this richer data makes a huge difference to the accuracy of our categorization model.

## Evaluation

We train our model with the updated training data from the previous section. We will refer to our model trained using the added data as the "updated model" henceforth. We compare our updated and original models according to both their categorization accuracy and how the improved accuracies help Deepseek-math in problem-solving.

### Accuracy Improvements

We first compare the accuracies of our original model and the updated model using a 25-question test set. The results are presented in Tables 3 and 4.

Here are the key takeaways from our results:

1. Our initial categorization model categorizes 64% of problems correctly, versus 84% of problems for our updated determination model.

2. Every problem that was correctly categorized by our initial model is also categorized correctly by our updated model.

3. On two of the four problems that our updated model got wrong, the probability of the correct category was close

| Problem | A | C | G | N | Answer | Correct Answer |
|---|---|---|---|---|---|---|
| 1 | 0.14 | 0.14 | 0.68 | 0.04 | G | G |
| 2 | 0.21 | 0.43 | 0.05 | 0.31 | C | C |
| 3 | 0.08 | 0.20 | 0.04 | 0.68 | N | C |
| 4 | 0.75 | 0.05 | 0.11 | 0.09 | A | A |
| 5 | 0.18 | 0.12 | 0.06 | 0.63 | N | N |
| 6 | 0.07 | 0.36 | 0.08 | 0.48 | N | A |
| 7 | 0.08 | 0.24 | 0.18 | 0.51 | N | C |
| 8 | 0.23 | 0.18 | 0.52 | 0.08 | G | G |
| 9 | 0.15 | 0.04 | 0.65 | 0.16 | G | G |
| 10 | 0.19 | 0.11 | 0.04 | 0.66 | N | A |
| 11 | 0.15 | 0.07 | 0.63 | 0.15 | G | G |
| 12 | 0.60 | 0.04 | 0.03 | 0.33 | A | A |
| 13 | 0.22 | 0.23 | 0.06 | 0.49 | N | N |
| 14 | 0.24 | 0.20 | 0.05 | 0.51 | N | N |
| 15 | 0.16 | 0.45 | 0.19 | 0.21 | C | C |
| 16 | 0.30 | 0.04 | 0.24 | 0.41 | N | G |
| 17 | 0.31 | 0.27 | 0.07 | 0.35 | N | C |
| 18 | 0.16 | 0.24 | 0.14 | 0.46 | N | G |
| 19 | 0.22 | 0.29 | 0.04 | 0.45 | N | C |
| 20 | 0.45 | 0.07 | 0.04 | 0.44 | A | N |
| 21 | 0.62 | 0.03 | 0.04 | 0.31 | A | A |
| 22 | 0.41 | 0.08 | 0.05 | 0.46 | N | N |
| 23 | 0.10 | 0.10 | 0.71 | 0.10 | G | G |
| 24 | 0.66 | 0.03 | 0.11 | 0.20 | A | A |
| 25 | 0.14 | 0.12 | 0.57 | 0.17 | G | G |

Table 3: Normalized probabilities for each category are shown in columns 2-5. The model picks the highest-probability category (column 6) and the correct category is shown in column 7. Incorrectly categorized problems are shown in orange or red. In particular, the red problems are incorrectly classified by our original model but correctly categorized by our updated model (Table 4)

| Problem | A | C | G | N | Answer | Correct Answer |
|---|---|---|---|---|---|---|
| 1 | 0.15 | 0.08 | 0.74 | 0.03 | G | G |
| 2 | 0.29 | 0.56 | 0.00 | 0.15 | C | C |
| 3 | 0.06 | 0.08 | 0.01 | 0.85 | N | C |
| 4 | 0.87 | 0.06 | 0.04 | 0.04 | A | A |
| 5 | 0.16 | 0.05 | 0.02 | 0.77 | N | N |
| 6 | 0.07 | 0.39 | 0.02 | 0.51 | N | A |
| 7 | 0.14 | 0.67 | 0.08 | 0.11 | C | C |
| 8 | 0.22 | 0.33 | 0.43 | 0.03 | G | G |
| 9 | 0.18 | 0.05 | 0.72 | 0.05 | G | G |
| 10 | 0.50 | 0.09 | 0.08 | 0.34 | A | A |
| 11 | 0.03 | 0.08 | 0.82 | 0.07 | G | G |
| 12 | 0.56 | 0.04 | 0.03 | 0.36 | A | A |
| 13 | 0.04 | 0.10 | 0.02 | 0.84 | N | N |
| 14 | 0.05 | 0.29 | 0.01 | 0.65 | N | N |
| 15 | 0.04 | 0.83 | 0.05 | 0.08 | C | C |
| 16 | 0.11 | 0.06 | 0.72 | 0.12 | G | G |
| 17 | 0.30 | 0.26 | 0.01 | 0.43 | N | C |
| 18 | 0.03 | 0.33 | 0.31 | 0.34 | N | G |
| 19 | 0.04 | 0.60 | 0.01 | 0.35 | C | C |
| 20 | 0.32 | 0.11 | 0.03 | 0.55 | N | N |
| 21 | 0.82 | 0.09 | 0.01 | 0.07 | A | A |
| 22 | 0.08 | 0.11 | 0.02 | 0.80 | N | N |
| 23 | 0.03 | 0.07 | 0.82 | 0.08 | G | G |
| 24 | 0.67 | 0.04 | 0.04 | 0.25 | A | A |
| 25 | 0.04 | 0.07 | 0.82 | 0.08 | G | G |

Table 4: Normalized probabilities for each category are shown in columns 2-5. The model picks the highest-probability category (column 6) and the correct category is shown in column 7. Incorrectly categorized problems are shown in orange.

to (but lower than) the category with the highest probability.

4. Our updated model got four problems wrong, each of which was incorrectly categorized as number theory. This shows the shortcomings of our model: even though fixing our dataset decreased the bias towards number theory, it still exists. (The initial model incorrectly categorized 9 problems, of which 8 were categorized as number theory.)

As can be seen, despite the shortcomings, the updated model presents a significant improvement. In Section , we study how this accuracy improvement translates to improvements in the "downstream" task of using Deepseek-math to solve problems. Before that, we consider the four incorrectly-categorized problems and discuss ideas for further improving our accuracy.

**On Further Improving Categorization.** Here are the problems that both models got wrong during the categorization test:

**Problem 5** *Let the 'sparkle' operation on positive integer $n$ consist of calculating the sum of the digits of $n$ and taking its*

factorial, e.g. the sparkle of 13 is $4! = 24$. A robot starts with a positive integer on a blackboard, then after each second for the rest of eternity, replaces the number on the board with its sparkle. For some 'special' numbers, if they're the first number, then eventually every number that appears will be less than 6. How many such special numbers are there with at most 36 digits?

**Problem 6** *For how many positive integers $m$ does the equation*

$$||x - 1| - 2| = \frac{m}{100}$$

*have $4$ distinct solutions?*

**Problem 7 (AIME 2020/7)** *A club consisting of $11$ men and $12$ women needs to choose a committee from among its members so that the number of women on the committee is one more than the number of men on the committee. The committee could have as few as $1$ member or as many as $23$ members. Let $N$ be the number of such committees that can be formed. Find the sum of the prime numbers that divide $N$.*

**Problem 8 (AIME 2020/8)** *A bug walks all day and sleeps all night. On the first day, it starts at point $O$, faces east, and walks a distance of $5$ units due east. Each night the bug rotates $60°$ counterclockwise. Each day it walks in this new*

| Problem | Outputs | Frequencies | Correct Answer | Correct Answer Frequencies | Correct? |
|---------|---------|-------------|----------------|---------------------------|----------|
| 1  | 2   | 1 | 52  | 0 | No  |
| 2  | 39  | 1 | 250 | 0 | No  |
| 3  | 1   | 2 | 702 | 0 | No  |
| 4  | 8   | 2 | 800 | 1 | No  |
| 5  | 310 | 3 | 211 | 0 | No  |
| 6  | 199 | 2 | 199 | 2 | Yes |
| 7  | 1   | 2 | 185 | 0 | No  |
| 8  | 100 | 3 | 320 | 0 | No  |
| 9  | 437 | 2 | 480 | 0 | No  |
| 10 | 100 | 2 | 199 | 1 | No  |
| 11 | 781 | 2 | 722 | 0 | No  |
| 12 | 139 | 2 | 139 | 2 | Yes |
| 13 | 2   | 3 | 307 | 1 | No  |
| 14 | 990 | 2 | 507 | 0 | No  |
| 15 | 4   | 1 | 341 | 0 | No  |
| 16 | 24  | 2 | 58  | 0 | No  |
| 17 | 396 | 4 | 539 | 0 | No  |
| 18 | 695 | 2 | 695 | 2 | Yes |
| 19 | 494 | 2 | 494 | 2 | Yes |
| 20 | 12  | 2 | 72  | 0 | No  |
| 21 | 32  | 2 | 108 | 0 | No  |
| 22 | 15  | 2 | 431 | 0 | No  |
| 23 | 91  | 1 | 91  | 1 | Yes |
| 24 | 158 | 1 | 483 | 0 | No  |
| 25 | 25  | 2 | 53  | 0 | No  |

Table 5: Outputs of Deepseek-math while using model-based choice between CT and PT. The second and third columns are the most frequent outputs for each problem.

*direction half as far as it walked the previous day. The bug gets arbitrarily close to the point P. Then $OP^2 = \frac{m}{n}$, where m and n are relatively prime positive integers. Find $m + n$.*

Both models predicted all of these questions to be number theory, whereas they should have been categorized as, respectively, combinatorics, algebra, combinatorics, and geometry.

Additionally, the errors in the third and fourth problems seem to be in the answer extraction. For example, the third problem states "Find the sum of the prime numbers that divide $N$", and the fourth problem states "relatively prime positive integers" (this is again the most common answer extraction). These are answer extraction errors that we specifically tried to fix in Section . This shows that our model is not perfect, and there is still a lot of room for improvement. One of the simplest ways to improve the model would be to add more problems from the AIME to its training dataset. Only 90 problems were added in Section , but adding more would lead to a much higher accuracy.

**Downstream Task Evaluation**

We utilize the same testing set and number of iterations as Section , but the categorization of problems will now be done *automatically* by the updated model we evaluated in the previous section rather than manually. This categorization is not perfect, so we should expect slightly worse results than problem-solving with perfect categorization.

1. The model answers 20% of the problems correctly, and has a nonzero correct answer frequency on 32% of the problems. These values are slightly worse than the values for perfect categorization (28%, 36%), and significantly better than the values for random choice (12%, 24%).

2. Surprisingly, we find that Deepseek-math solves problem 23 in this test set, which is not only a very difficult problem (it was problem 13 on the 2015 AIME), but was also unsolved even with perfect categorization.

As can be seen in Table 5, our model answers five problems correctly out of the 25, and gets the correct answer on three other problems, though the correct answer in that case did not have the highest frequency. This is a very significant improvement from randomly choosing between CT and PT, but there is still a gap between the accuracy here and accuracy with perfect categorization.

**Conclusion**

In this paper, we showed that problem categorization could be used to facilitate math problem-solving by LLMs. This is because, with some guidance on what category a problem belongs to, we can provide the LLM with a tactic in {CT, PT} that is *likelier to work on the specific problem*. This provides a large improvement over other approaches.

We find that the accuracy of LLMs when a strategy is chosen using perfect categorization is much higher than the

accuracy of LLMs when a random strategy is chosen. The accuracy of LLMs when a strategy is chosen using our categorization model lies in between these two.

Our model achieves around 84% accuracy in predicting the categories of problems. While this is fairly high, the accuracy can likely be further improved by using a larger and even more accurate dataset.

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
