# OpenReview forum: "Improving Math Problem Solving in Large Language Models Through Categorization and Strategy Tailoring"
_AAAI.org/2025/Workshop/NeurMAD — AAAI 2025 Workshop NeurMAD Submission_

### Official Review · Reviewer_hBvu · 2024-12-20
**A weak categorization between chain-of-thought and program-of-thought**

**Rating:** 4
**Confidence:** 4

**Review:**

This paper proposes to solve math problems by automatically selecting between chain-of-thought (CT) and program-of-though (PT) prompting. The selection is implemented with a categorization model using a 3-layer neural network based on word frequency features.

Despite some interesting insights, the paper has some major weaknesses:

* Weak categorization model. The categorization model is based on word frequency features. As mentioned in the paper, one of the main challenges encountered in categorization is ‘answer extraction,’ which is hard to address with word frequency features. Hence, a stronger sequence model (such as a Transformer) should be taken into account for solving the categorization.
* Why not compute the posterior probability? The current approach takes the argmax of the output distribution of the categorization model, and based on the selected category samples from the P_s(. | c), where s in {CT, PT} and c in {algebra, geometry, number theory, combinatorics}. However, one could also incorporate the output distribution of the categorization model into the computation of the posterior sampling distribution.
Moreover, the definition of the prior sampling distribution {CT, PT} is somewhat arbitrary and not justified.
* It is not clear how the approach is helping to reduce hallucinations in LLMs. If the approach would indeed reduce hallucinations, it should be demonstrated empirically.
* The methodology is not clear: the paper is missing important details about network architecture for the categorization model, example prompts, etc. In its current form, the experimental results of this paper are not reproducible.

Minor weaknesses/corrections
* References of AlphaProof and Alphageometry link to news articles instead of the original papers.
* The section references are broken.

---

### Official Review · Reviewer_nY3f · 2024-12-29
**No details of subjectwise intructions used, Weak problem categorization**

**Rating:** 5
**Confidence:** 3

**Review:**

The paper proposes a novel method to solve mathematical problems by classifying them into categories and strategies.

Key issues:

- Hallucination was mentioned earlier in the paper, but no clear metrics were used to show if the approach reduces hallucination.
- It's not clear what category-wise instructions were given to the model to solve problems.
- The categorization of problems was weak; a more robust approach could have been to use a small transformer-based model for category classification.
- A transformer model trained to determine which strategy is better for a particular problem could have been more effective instead of using a pre-defined distribution.
- Could have provided details of the prompt structure used to convey problems,categories and instructions to the model.

Minor issues:

- Tables could have been more readable.

---

### Decision · Program_Chairs · 2024-12-30

**Decision:**

Reject

**Comment:**

 We agree with the opinions of  the reviewers.